# NON-LINEAR REWARDS FOR SUCCESSOR FEATURES

## ABSTRACT

Reinforcement Learning algorithms have reached new heights in performance, often overtaking humans on several challenging tasks such as Atari and Go. However, the resulting models typically learn fragile policies that are unable to transfer between tasks without full retraining. *Successor features* aim to improve this situation by decomposing the policy into two components: one capturing environmental dynamics and the other modelling reward. Under this framework, transfer between related tasks requires only training the reward component. However, *successor features* builds upon the assumption that the current reward can be predicted from a linear combination of state features; an assumption with no guarantee. This paper proposes a novel improvement to the *successor feature* framework, where we instead assume that the reward function is a non-linear function of the state features, thereby increasing its representational power. After derivation of the new state-action value function, the decomposition includes a second term that learns the auto-correlation matrix between state features. Experimentally, we show this term explicitly models the environment's stochasticity and can also be used in place of $\epsilon$-greedy exploration methods during transfer. The performance of the proposed improvements to the *successor feature* framework is validated empirically on navigation tasks and control of a simulated robotic arm.

Recently, Reinforcement Learning (RL) algorithms have achieved superhuman performance in several challenging domains, such as Atari (Mnih et al., 2015), Go (Silver et al., 2016), and Starcraft II (Vinyals et al., 2019). The main driver of these successes has been the use of deep neural networks, which are a class of powerful non-linear function approximators, with RL algorithms (LeCun et al., 2015). However, this class of Deep Reinforcement Learning (Deep RL) algorithms require immense amounts of data within an environment, often ranging from tens to hundreds of millions of samples (Arulkumaran et al., 2017). Furthermore, commonly used algorithms often have difficulty in transferring a learned policy between related tasks, such as where the environmental dynamics remain constant, but the goal changes. In this case, the model must either be retrained completely or fine-tuned on the new task, in both cases requiring millions of additional samples. If the state dynamics are constant, but the reward structure varies between tasks, it is wasteful to retrain the entire model.

A more pragmatic approach would be to decompose the RL agent's policy such that separate functions can learn the state dynamics and the reward structure; doing so enables reuse of the dynamics model and only requires learning the reward component. *Successor features* (Dayan, 1993) do precisely this; a model-free policy's action-value function is expressed as the dot product between a vector of expected discounted future state occupancies, the successor features, and another vector representing the immediate reward in each of those successor states. The factorization follows from the assumption that reward can be predicted as the dot product between a state representation vector and a learned reward vector. Therefore, transfer to a new task requires relearning only the reward parameters instead of the entire model and amounts to the supervised learning problem of predicting the current state's immediate reward.

This factorization can be limiting because it is assumed that the reward is a linear function of the current state, which might not always be the case as the encoded features might not capture the required quantity for accurate reward modelling (Eysenbach et al., 2018; Hansen et al., 2019). Therefore, this paper introduces a new form for the reward function: non-linear with respect to the current state. We assume that the learned features are not optimal and the reward cannot be predicted directly from the raw features, which is not a strong assumption. This form increases the reward function's representational power and makes it possible to incorporate the current state into reward

estimation; lessening the burden on the encoder components. Under the new reward formulation, a secondary term emerges, which learns the future expected auto-correlation matrix of the state features. This new secondary term, referred to as $\Lambda$, can be exploited as a possible avenue for directed exploration. Exploring the environment using $\Lambda$ allows us to exploit and reuse learned environmental knowledge instead of relying on a purely random approach for exploration, such as $\epsilon$-greedy.

Following this, the contributions of this research are as follows:

- A novel formulation of successor features that uses a non-linear reward function. This formulation increases the representational power of the reward function.
- Under the new reward formulation, a second term appears that models the future expected auto-correlation matrix of the state features.
- We provide preliminary results that show the second term can be used for guided exploration during transfer instead of relying on $\epsilon$-greedy exploration.

After the introduction of relevant background material in Section 1, we introduce the successor feature framework with a non-linear reward function in Section 2, Section 3 provides experimental support and provides an analysis of the new term in the decomposition. The paper concludes with a final discussion and possible avenues for future work in Section 4.

## 1 BACKGROUND

### 1.1 REINFORCEMENT LEARNING

Consider the interaction between an agent and an environment modelled by a Markov decision process (MDP) (Puterman, 2014). An MDP is defined as a set of states $\mathcal{S}$, a set of actions $\mathcal{A}$, a reward function $R : S \rightarrow \mathbb{R}$, a discount factor $\gamma \in [0, 1]$, and a transition function $T : \mathcal{S} \times \mathcal{A} \rightarrow [0, 1]$. The transition function gives the next-state distribution upon taking action $a$ in state $s$ and is often referred to as the dynamics of the MDP.

The objective of the agent in RL is to find a policy $\pi$, a mapping from states to actions, which maximizes the expected discounted sum of rewards within the environment. One solution to this problem is to rely on learning a value function, where the *action-value function* of a policy $\pi$ is defined as:

$$Q^\pi(s,a) = \mathbb{E}^\pi \left[ \sum_{t=0}^\infty \gamma^t R(s_t) | S_t = s, A_t = a \right]$$

where $\mathbb{E}^\pi[\dots]$ denotes the expected value when following the policy $\pi$. The policy is learned using an alternating process of *policy evaluation*, given the action-value of a particular policy and *policy improvement*, which derives a new policy that is *greedy* with respect to $Q^\pi(s,a)$ (Puterman, 2014).

### 1.2 SUCCESSOR FEATURES

Successor Features (SF) offer a decomposition of the Q-value function and have been mentioned under various names and interpretations (Dayan, 1993; Kulkarni et al., 2016; Barreto et al., 2017; Machado et al., 2017). This decomposition follows from the assumption that the reward function can be approximately represented as a linear combination of learned features $\phi(s; \theta_\phi)$ extracted by a neural network with parameters $\theta_\phi$ and a reward weight vector $w$. As such, the expected one-step reward can be computed as: $r(s,a) = \phi(s; \theta_\phi)^\top w$. Following from this, the Q function can be rewritten as:

$$Q(s,a) \approx \mathbb{E}^\pi \left[ r_{t+1} + \gamma r_{t+2} + \dots | S_t = s, A_t = a \right]$$

$$= \mathbb{E}^\pi \left[ \phi(s_{t+1}; \theta_\phi)^\top w + \phi(s_{t+2}; \theta_\phi)^\top w + \dots | S_t = s, A_t = a \right]$$

$$Q(s,a) = \psi^\pi(s,a)^\top \cdot w$$

where $\psi(s,a)$ are referred to as the *successor features* under policy $\pi$. The $i^{\text{th}}$ component of $\psi(s,a)$ provides the expected discounted sum of $\phi_t^{(i)}$ when following policy $\pi$ starting from state $s$ and

action $a$. It is assumed that the features $\phi(s; \theta_\phi)$ are representative of the state $s$, such that $\psi(.)$ can be turned into a function $\psi^\pi(\phi(s_t; \theta_\phi), a_t)$. For brevity, $\phi(s_t; \theta_\phi)$ is referred to simply as $\phi_t$ and $\psi^\pi(s, a)$ as $\psi(s, a)$.

The decomposition neatly separates the Q-function into two learning problems, for $\psi^\pi$ and $w$: estimating the features under the current policy dynamics, and estimating the reward given a state. Because the decomposition still has the same form as the Q-function, the successor features are computed using a Bellman equation update in which the reward function is replaced by $\phi_t$:

$$\psi^\pi(\phi_t, a_t) = \phi_t + \gamma \mathbb{E}\left[\psi^\pi(\phi_{t+1}, a_{t+1})\right]$$

such that approximate successor features can be learned using an RL method, such as Q-Learning (Szepesvári, 2009).

Following from this, the approximation of the reward vector $w$ becomes a supervised learning problem. Often, this weight is learned using ordinary least squares from the sampled environmental data. One benefit of having a decoupled representation is that only the relevant function must be relearned when either the dynamics or the reward changes. Therefore, if the task changes, but the environmental dynamics remain constant, only the reward vector parameters $w$ must be relearned, which are minimal compared to the total number of parameters in the full model.

## 2 Model, Architecture, and Training

The Successor Feature framework has several limitations, primarily stemming from the assumptions around its derivation, such as constant environmental structure between tasks or that the reward can be linearly predicted from state features. Work towards solving the former has been developed by Zhang et al. (2017) whereby they learn a linear mapping between task state features. The latter assumption, whereby the reward is assumed to be a linear mapping of state features, is not guaranteed and, as we show, the Successor Feature framework fails in such cases. Therefore, the method presented in this section aims to provide a stronger guarantee of the framework's performance in such cases by developing a more robust reward component.

This section discusses our change to the successor feature framework, which adjusts reward function, from a linear function, to a non-linear function. First, a discussion of the new decomposition is given with the full derivation provided in Appendix A. Then experimental support for this change will be presented and analyzed to examine what the new term in the decomposition learns.

### 2.1 Non-linear Reward Function

The successor feature framework builds upon the assumption that the current reward $r_t$ can be represented by the linear combination of the current state representation $\phi_t \in \mathbb{R}^z$ and a learned reward vector $w \in \mathbb{R}^z$, such that $r_t = \phi_t^\top w$. This form is limiting because there is no guarantee that the reward will be a linear combination of the state features or that the required state features can be learned by the encoder (Eysenbach et al., 2018; Hansen et al., 2019). In practice the optimal state features are often not learned; therefore, we build on the basis that the state features are sub-optimal, which in itself is not a strong assumption. To increase the flexibility of this reward model, let us consider the following form:

$$r_t = \phi_t^\top \mathbf{o} + \phi_t^\top \mathbf{A} \phi_t \tag{1}$$

where $\{\phi_t, \mathbf{o}\} \in \mathbb{R}^z$, and $\mathbf{A} \in \mathbb{R}^{z \times z}$. Both $\mathbf{o}$ and $\mathbf{A}$ are learnable parameters modelling the reward structure of the environment. Equation 1 shows that the formulation introduces a non-linear transformation with respect to $\phi$. Comparing this with the original formulation, it is evident that this is equivalent to setting $w = \mathbf{o} + \mathbf{A}\phi$. The state-action value function $Q(s, a)$, under this new reward structure, can be derived to yield:

$$Q^\pi(s_t, a) = \psi^\pi(s_t, a)^\top \mathbf{o} + \beta \mathbf{tr}(\mathbf{A}\Lambda^\pi(s_t, a)) \tag{2}$$

where $\beta \in \{0, 1\}$ controls the inclusion of $\Lambda$ and $\mathbf{tr}$ is the trace operator. It can now be shown that $\psi$ and $\Lambda$ satisfy the Bellman equation (Bellman, 1966):

$$\psi^\pi(s_t, a) = \mathbb{E}^\pi[\phi_{t+1} + \gamma \psi(s_{t+1}, \pi(s_{t+1}))|S_t = s, A_t = a] \tag{3}$$

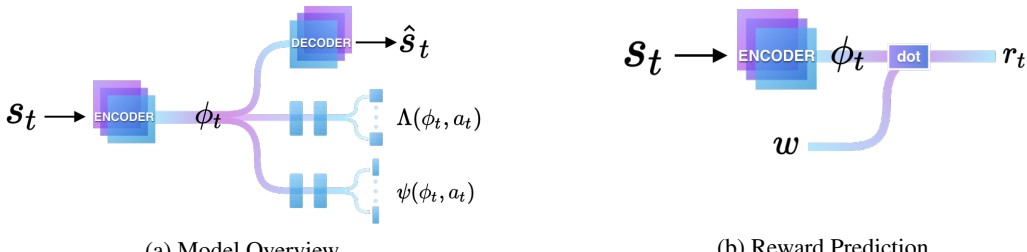

(a) Model Overview            (b) Reward Prediction

Figure 1: **Model Overview** a) The encoder transforms the raw state into an internal state representation $\phi_t$. The state representation $\phi_t$ is used by the decoder, $\Lambda(\cdot, a_t)$, and $\psi(\cdot, a_t)$. The decoder tries to reconstruct the raw input $s_t$ from the state representation $\phi_t$. $\Lambda$ and $\psi$ produce one output per action, with the former predicting matrices and the latter predicting vectors. Also, both $\Lambda$ and $\psi$ use two hidden layers to process $\phi_t$ before their final layers. b) Reward prediction by dotting the current state $\phi_t$, produced by the encoder, and reward weight $w$.

$$\Lambda^{\pi}(s_t, a) = \mathbb{E}^{\pi}[\phi_{t+1}\phi_{t+1}^{\top} + \gamma\Lambda(s_{t+1}, \pi(s_{t+1}))|S_t = s, A_t = a] \tag{4}$$

where for $\psi$ and $\Lambda$, $\phi$ and $\phi\phi^{\top}$ respectively play the role of rewards. In addition to $\psi$, it is now necessary to model $\Lambda$, which outputs an $\mathbb{R}^{z \times z}$ matrix *per* action. The quantity $\phi_t\phi_t^{\top}$ can be interpreted as an auto-correlation matrix of the state features. We can see that this form allows the $\Lambda$ term to model some form of future expected stochasticity of the environment. For example, the diagonal of $\Lambda$ will model a second order moment capturing each feature's change with respect to itself $\phi^1$. We provide analysis and further discussion of $\Lambda$ in Section 3.5.

## 2.2 Model Structure and Training

The proposed model, shown in Figure 1a, uses an encoder to produce a state embedding $\phi_t$ consumed by downstream modelling tasks. Figure 1b shows how the current reward $r_t$ is predicted using $w$, with $w = \mathbf{o} + \mathbf{A}\phi$, and current state representation $\phi_t$; this process is defined in Equation 1. Similar to previous work with successor features, the structure includes pathways for an encode-decode task and successor feature prediction $\psi$ (Machado et al., 2017; Kulkarni et al., 2016; Zhang et al., 2017). The decoder network ensures that the features learned by the encoder, which produces $\phi$, contain useful information for prediction. Furthermore, only the gradients from the state-dependent and reward prediction tasks modify the encoder parameters, and therefore $\phi$. An additional branch is added, by way of the non-linear reward function, to model the quantity $\Lambda(s, a)$. This branch's output is a matrix, which differs from the vector predicting branches $\psi$ and $\phi$.

The encode-decode task is trained by minimizing the mean squared difference between the input $s_t$ and the decoder's reconstructed version $\hat{x}_t$ from $\phi$:

$$\mathcal{L}^d(s_t; \theta^{\phi}, \hat{\theta}^{\phi}) = [s_t - g(\phi_t; \hat{\theta}^{\phi})]^2 \tag{5}$$

where $\phi$ is the output of the encoder with parameters $\theta^{\phi}$ and $g(\cdot; \hat{\theta}^{\phi})$ produces the output of the decoder with parameters $\hat{\theta}^{\phi}$. As mentioned previously, we train $\psi$ and $\Lambda$, parameterized with $\theta^{\psi}$ and $\theta^{\Lambda}$ respectively, using the Bellman equations to minimize the following losses:

$$\mathcal{L}^{\psi}(s_t, a_t; \theta^{\psi}) = \mathbb{E}[(\phi_t^- + \gamma\psi(s_{t+1}, a^*; \theta^{-\psi}) - \psi(s_t, a_t; \theta^{\psi}))^2] \tag{6}$$

$$\mathcal{L}^{\Lambda}(s_t, a_t; \theta^{\Lambda}) = \mathbb{E}[(\phi_t^-\phi_t^{-\top} + \gamma\Lambda(s_{t+1}, a^*; \theta^{-\Lambda}) - \Lambda(s_t, a_t; \theta^{\Lambda}))^2] \tag{7}$$

where $a^* = max_{a^*}Q(s, a^*)$. To help stabilize learning, we use lagged versions of $\theta^{\phi}$, $\theta^{\psi}$, and $\theta^{\Lambda}$ as done by Mnih et al. (2015); the lagged version is signified with the $-$ symbol in the exponent.

Unfortunately, as the dimensionality of $z$ grows, the number of parameters needed by $\Lambda$ grows quadratically. However, by identifying the $\phi_t\phi_t^{\top}$ term in $\Lambda(s, a)$ as a symmetric matrix, it is possible to model only the upper triangular portion of the matrix[2], requiring about half the number of

---

[1] A quantity close to the variance, but not zero mean.

[2] It would still be necessary to manipulate this matrix so that it forms a full matrix.

parameters. To further reduce parameters, each $\psi$ and $\Lambda$ pathways have two hidden layers before their outputs, reflected in Figure 1a. In this way, the parameters are shared amongst pathways, which contrasts with other works with multiple sets of layers per action $a \in \mathcal{A}$ (Kulkarni et al., 2016; Zhang et al., 2017). To learn the reward parameters $\mathbf{A}$ and $\mathbf{o}$, which are the parameters of the approximated non-linear reward function, the following squared loss function is used:

$$\mathcal{L}^r(s_t; o, A) = [r_t - \phi_t^\top \mathbf{o} + \beta \phi_t^\top \mathbf{A} \phi_t]^2 \tag{8}$$

We found that the loss $\mathcal{L}^r$ is not enough to train the parameters $\mathbf{A}$ and $\mathbf{o}$ alone, and that also regressing towards the Q-function target to be more informative. Therefore, similar to Ma et al. (2020) we use the additional loss:

$$\mathcal{L}^Q(s_t, a_t; o, A, \theta^\psi, \theta\Lambda) = [\hat{Q}_t - \psi(s_t, a_t; \theta^\psi)^\top o + \beta \mathbf{tr}(\mathbf{A}\Lambda(s_t, a_t; \theta^\Lambda))]^2 \tag{9}$$

where $\hat{Q}_t = r_t + \gamma Q(s_{t+1}, a^*)$ is the target term[3]. The additional $\mathcal{L}^Q$ loss can be treated as an auxiliary loss that forces the agent to learn values relevant to the actual quantity used for decision making. The $\mathcal{L}^Q$ loss is additively included and scaled with a hyperparameter $\lambda$, which, in this work, is typically set between $0.01$ and $0.1$. It does not adjust the feature parameters involved in the prediction of $\phi_t$.

However, similar to $\Lambda$, as the dimensionality of $z$ increases, so does the number of parameters needed for modelling matrix $\mathbf{A} \in \mathbb{R}^{z \times z}$. Therefore, in the interest of reducing the number of parameters we use a factorization that splits the matrix $\mathbf{A} \in \mathbb{R}^{z \times z}$ into two parts with a smaller inner dimension $f$, $\mathbf{A} = \mathbf{L} \cdot \mathbf{R}^\top$, where $\{\mathbf{L}, \mathbf{R}\} \in \mathbb{R}^{z \times f}$. By factoring the matrix in this way, we require $2 \times z \times f$ parameters instead of $z \times z$. If we use values for $f$ smaller than $\frac{z}{2}$, we reduce the number of parameters required by matrix $\mathbf{A}$. A similar factorization was suggested in the context of visual question answering (Yu et al., 2017; Fukui et al., 2016). The factorization of $\mathbf{A}$ was primarily done to reduce the total number of parameters in our model. In Appendix E we show that using the full matrix is still a tractable learning problem and differs only slightly in overall performance. Combining our losses, the composite loss function is the sum of the four losses given above:

$$\mathcal{L}(\theta^\phi, \hat{\theta}^\phi, \theta^\psi, \theta^\Lambda, o, A) = \mathcal{L}^d + \mathcal{L}^\psi + \beta \mathcal{L}^\Lambda + \mathcal{L}^r + \lambda \mathcal{L}^Q \tag{10}$$

In practice, to optimize Equation 10 with respect to its parameters, $(\theta^\psi, \theta^\Lambda)$ and $(\theta^\phi, \hat{\theta}^\phi, o, \mathbf{A})$ are iteratively updated. Doing so increases the stability of the approximations learned by the model and ensures that the branches modelling $\psi$ and $\Lambda$ do not backpropagate gradients to affect $\theta^\phi$ (Machado et al., 2017; Zhang et al., 2017; Kulkarni et al., 2016). Additionally, by training in this way, the state representation $\phi$ can learn features that are both a good predictor of the reward $r_t$ and useful in discriminating between states (Kulkarni et al., 2016).

## 3 EXPERIMENTS

This section examines the properties of the proposed approach on Axes, a navigation task, and on Reacher, a robotic control task built using the MuJoCo engine (Todorov et al., 2012). The environments are shown in Figure 2; they each contain several tasks specified by goal location and are split between training and test distributions. The experiments examine what the model learns and give a preliminary examination of possible uses for the $\Lambda$ component in the proposed model.

Specifically, we show that the non-linear assumption increases the representational power of the reward function, provide evidence that the model can learn non-linear rewards, examine the learned $\Lambda$ function to understand if it captures environmental stochasticity, and evaluate different guided exploration strategies using the $\Lambda$ term on new tasks. Additional environment details are provided in Appendix B.

---

[3]This term is written to be generally applicable to different training methodologies. As we train with an A3C-type framework, we regress towards an estimate of the return from the environment trajectories.

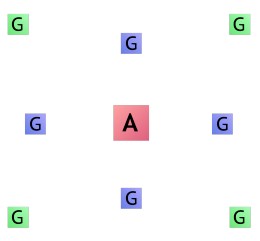
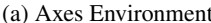
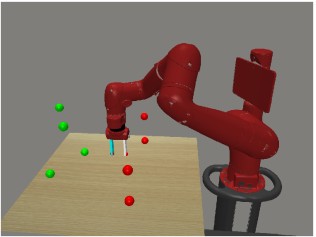

|                          |                          |
| :----------------------: | :----------------------: |
| (a) Axes Environment.    | (b) Reacher Environment. |

Figure 2: **Environments** a) A graphical representation of the Axes environment. The agent, shown as a red square, must traverse to various goal locations marked with the letter "G". The eight goal locations are split between training, shown as blue boxes, and testing, shown as green boxes. b) A rendering of the Reacher task. The agent controls the robotic Sawyer arm to move the end-effector to a 3D point in space. The eight goal locations are shown as balls. Training goals as green, and test goals as red.

## 3.1 ENVIRONMENTS

**Axes**: In this environment, shown in Figure 2a, the agent, shown by the red square, must traverse the map to reach a goal location using four actions: *up*, *down*, *left*, and *right*. The agent receives a reward equal to the negative distance between itself and the target goal at each step. The agent's state space contains the $(x, y)$ location of itself and the current target goal within the environment such that the state $s_t \in \mathbb{R}^4$. With this state space the agent must learn a reward function that can approximate the distance between itself and the goal location, $d(a, b) = \sqrt{(b_x - a_x)^2 + (b_y - a_y)^2}$, a non-linear function. We expect that the linear variant will fail in this task as it cannot model a non-linear reward function. This experiment shows that the non-linear reward function can compensate for a weak state representation.

**Reacher:** The second environment is a control task defined in the MuJoCo physics engine (Todorov et al., 2012), shown in Figure 2b. A modified version of the robotic model provided by Metaworld (Yu et al., 2019) was used. This environment was chosen to show that the proposed method can scale to difficult control tasks. In this environment, the agent must move a simulated robotic arm to a specific 3D point in space by activating four torque controlled motors. Similarly to Axes, the environment has predefined tasks split between training and test distributions, with the eight goals shown in Figure 2b as green and red balls, respectively. The state space is represented as the $(x, y, z)$ location of the current goal and the end effector, such that $s \in \mathbb{R}^6$. Reacher is similar to the Axes environment, in that the agent must compute the distance between the two points. The agent receives a reward equal to the negative distance between the end-effector and the current target goal at each step. We discretize the actions such that the agent has nine discrete actions that control the arms movements.

## 3.2 EXPERIMENTAL SETUP

The train and test setup was similar to those used in previous studies. The agent is trained on a distribution of randomly sampled train tasks; then, during testing, we change to a distribution of unseen tasks. A single policy $\pi$ is trained over all tasks. During transfer to unseen tasks, the model re-learns only the reward parameters, with the remainder of the model frozen (Zhang et al., 2017; Kulkarni et al., 2016). The newly learned policy will vary from the original but is able to exploit previously learned environment knowledge for the new tasks.

Our method is compared against a baseline version of the successor feature framework, most similar to that of Kulkarni et al. (2016). This baseline is identical in all ways to the proposed method except for the exclusion of terms containing $\Lambda$, specifically Equations 2 and 10, which can be obtained by setting $\beta = 0$. A uniform random action baseline was considered in all environments to act as a floor.

The number of fully connected layers within the encoder of each model varied between environments. The Axes environment used one layer while the Reacher environment used two layers in their encoders. Initially, on the Axes environment, the models all used the raw features with no encoder

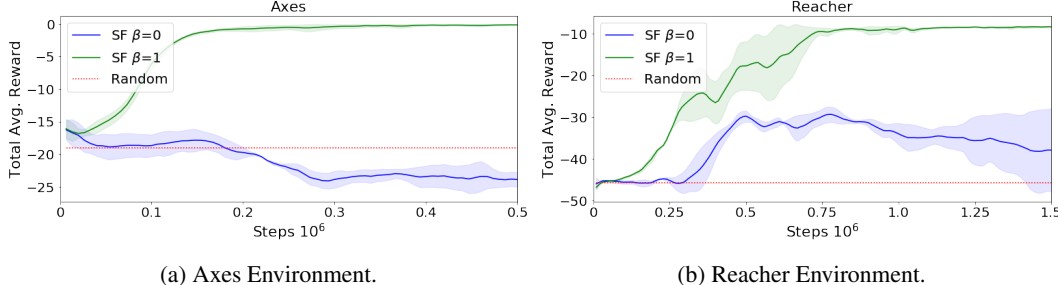

(a) Axes Environment.                               (b) Reacher Environment.

Figure 3: Performance in Axes and Reacher environments. The propose model, with the non-linear reward, is shown as the green line, while the linear variant is shown as the blue line. The non-linear variant outperforms the linear variant while differing by only the inclusion of the $\Lambda$ component, controlled with the $\beta$ hyperparameter. We assume the worse than random performance of the linear variant is due to the model learning a degenerate state representation and therefore policy.

such that $\phi_t = \mathbb{I}(s_t)$. We found this led to worse performance for the linear model as it now had no chance to learn a suitable encoding of the features for reward prediction. In Appendix E, we performed an ablation within the Axes environment that controls for the number of parameters in both variants of the model; we found even with equal parameters there is little impact on overall performance.

We report each model's mean performance on all plots as the average over three runs with varied seeds. Each plot includes the standard deviation over all runs as a shaded area. Additional experiment and model details can be found in Appendix C.

### 3.3 ENVIRONMENT PERFORMANCE

The first step was to examine the performance of the proposed method against various baselines. The primary point of comparison was between the proposed method and the original formulation of the *successor feature* framework, which can be recovered exactly by setting $\beta = 0$ in Equations 2 and 10 of the proposed model. The result of these experiments are shown in Figure 3 for both Axes, on the left, and Reacher, on the right. As shown, all versions of the proposed model using the non-linear reward outperformed the other baseline methods in either convergence speed or overall performance. Clearly, using a non-linear reward function greatly improved flexibility and performance, enabling the proposed method to learn a more accurate reward model for the task.

The failure of the linear reward model, SF $\beta = 0$ of Figure 4a and 4b, in both environments is expected. The model, with the linear reward function, is not able to appropriately model the environments reward structure as the reward is a non-linear function of the state; in this case, the agent's coordinates and the current goal location. It was found that increasing the representational power of the encoder with additional fully connected layers helps but does not allow the model to match performance of the non-linear variant (see Appendix E). We also hypothesize that the poor performance of the linear model, leading it to eventually perform worse than random, is due to the model learning degenerate features.

### 3.4 TASK TRANSFER

An important property of the *successor feature* framework is the ability to adapt rapidly to new tasks within the same environment. Adaption, or transfer, is accomplished by freezing the model's state-dependent components, such as $\psi$, and learning only the reward parameters $w$. Because $w$ is often a small vector equal to the embedding dimensionality of $\phi$, it can be quickly learned. This section examines how well the proposed method, with the additional branch for $\Lambda$ and the reward parameter $\mathbf{A}$, can transfer to new tasks in both the Axes and Reacher environment.

After training the models to convergence, we change the task distribution within the environment. After the task distribution change, denoted by the dashed vertical line in Figure 4, all model parameters were frozen except those pertaining to the reward functions, which were then trained using samples

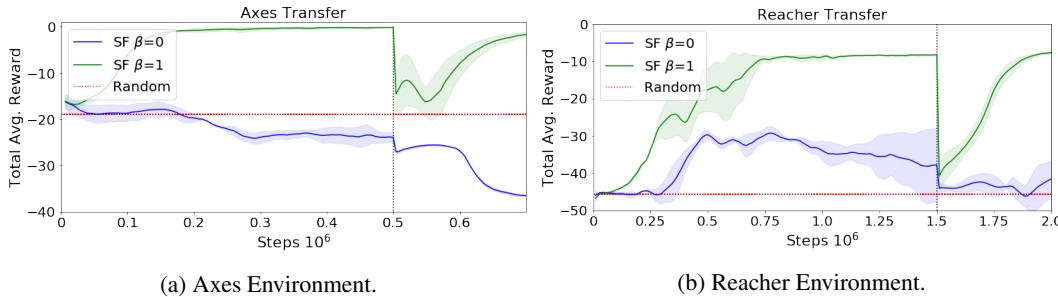

(a) Axes Environment.

(b) Reacher Environment.

Figure 4: Transfer Performance on the Axes and Reacher environments. Performance on the Axes environments is lower than training as the test tasks are on average farther from the agent's initial position.

from this new task. During transfer, we scaled the learning rate by a fixed factor, randomly initialize the learned weights, and re-decay the $\epsilon$ value so the method has a chance to explore. Full details are provided in Appendix D.

Figure 4 clearly shows that our proposed method converged quickly to the new test tasks and does so faster than training from scratch on both the Axes and Reacher environments. Faster learning indicates that the model is reusing previously learned task knowledge. We see that the linear variant is still unable to model the reward within the environment, leading to poor performance.

## 3.5 MODELLING ENVIRONMENTAL STOCHASTICITY

This section examines the $\Lambda$ function to determine whether it can capture stochasticity in the environment. We use the Axes environment as it is easy to examine the representations learned by the model. However, to do so, we modify the Axes environment to include some randomness. The modified version, referred to as *half-random*, is identical in all aspects to the base version except for a location-based conditional that affects the agent's actions. In other words, if the agent is within the positive $x$ quadrant of the map, $x > 0$, then actions are randomly perturbed with a fixed probability. Otherwise, they are fully deterministic. This randomness is shown in Figure 5a by the red shaded area.

After training to convergence on the half-random variant, we examine the $\Lambda$ function that the model has learned. Because the $\Lambda$ function is modelling the auto-correlation matrix, the future expected correlation of each feature with itself is found by looking along the diagonal. Figure 5b shows the result of plotting a diagonal value, in this case, feature 1, of the $\Lambda$-matrix over the entire state space. We can see that one of the diagonal components of $\Lambda$ did indeed learn to approximately model the conditional random field within the environment.

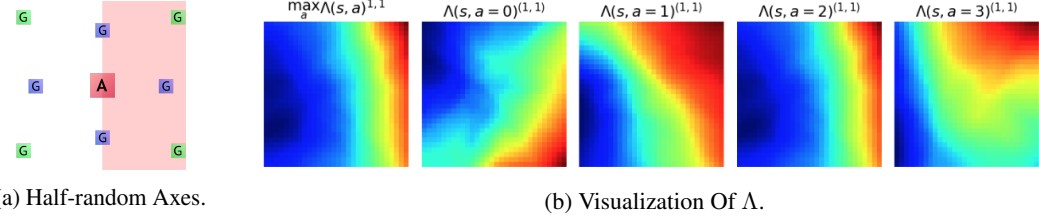

(a) Half-random Axes.

(b) Visualization Of $\Lambda$.

Figure 5: Visualization of Lambda Function on half-random Axes. a) The half-random variant of Axes. b) The learned expected future correlation of one feature with itself along $\Lambda$'s diagonal is visualized over the entire state space. The first column is the max value of $\Lambda$ over the actions. The remaining columns, from left to right, correspond to each action: *left*, *up*, *right*, and *down*. Red and blue correspond to maximal and minimal values.

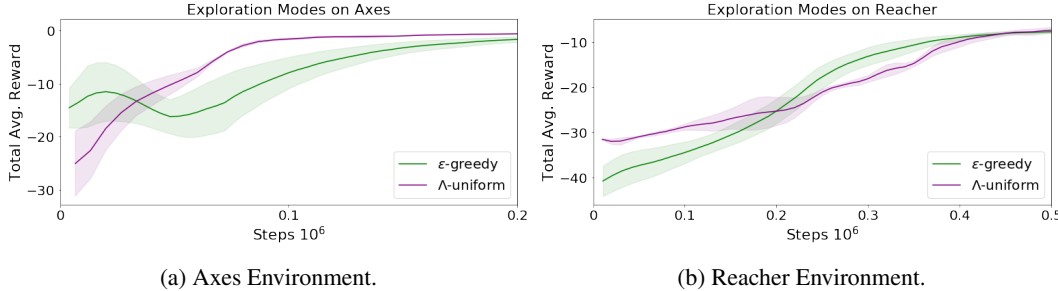

(a) Axes Environment.  (b) Reacher Environment.

Figure 6: Guided Exploration: The $\Lambda$ component of the proposed model is used to guide exploration during transfer. By using $\Lambda$ the agent explores in directions with large variance in the state space.

### 3.6 GUIDED EXPLORATION WITH $\Lambda$

This section presents preliminary results on possible uses for the $\Lambda$-function. Specifically, whether it is possible to use the $\Lambda$-function for guided exploration during transfer within the Axes and Reacher environments.

The Successor Features, given in Equation 2, can be interpreted as predicting the future expected path taken by the policy $\pi$ in an environment. Under this interpretation, $\psi$ can be seen as capturing the expected features of the states and $\Lambda$ the expected variance between state features along these pathways. Under an $\epsilon$-greedy policy, the captured variance would be induced by the random actions taken. Adding noise to the $\Lambda$ component would then perturb around the expected path, the one captured by $\psi$, to nearby states as done by a $\epsilon$-greedy policy. Therefore, instead of using $\epsilon$-greedy exploration, it is possible to add noise to $\Lambda$ during transfer, such that $\hat{\Lambda}(s, a) = \Lambda(s, a) + \epsilon \Lambda(s, a)$, where $\epsilon$ is sampled from some distribution. During learning, the variance of the sampling distribution, controlled by $\alpha$, can be annealed to some final value. The actions are then sampled from the model at time $t$ as:

$$a_t = argmax_{a^*} \left\{ \psi(s_t, a^*)^\top o + tr(\mathbf{A}\hat{\Lambda}(s_t, a^*)) \right\} \tag{11}$$

Various permutations of sampling distributions and structures were examined. Experimentally, it was found that using a scalar value sampled from uniform noise, that is $\epsilon \sim \mathcal{U}(-\alpha, \alpha)$ where $\epsilon \in [-\alpha, \alpha]$, provides the best performance. From Figure 6, we see that using $\Lambda$ for directed exploration is a viable alternative. In Figure 6a the directed exploration method manages to converge faster than $\epsilon$-greedy. We believe that using $\Lambda$ lets the model efficiently explore in directions along expected state pathways and reuses previously gained knowledge towards new tasks.

## 4 CONCLUSION & FUTURE WORK

In this paper, we have derived a novel formulation of successor features with a non-linear reward. We have shown that the agent can model reward non-linear reward structure, not possible under the old linear formulation. Further, we have shown the utility of the $\Lambda$ term that appears in the derivation of the new state-action function. Experimentally, we have shown that the $\Lambda$ term is able to capture the stochastic nature of an environment and can be used for directed exploration.

In future work, we aim to explore the $\Lambda$ function deeply. Specifically, what $\Lambda$ learns and if we can find a formulation that learns the future expected variance. Other possible avenues of future work include improvements to directed exploration with different annealing and whether our finding with the $\Lambda$ function can help improve the Q-function.

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

APPENDIX

# A  NON-LINEAR REWARD DERIVATION

Here we provide the derivation for the non-linear reward in the successor framework. First, we start by assuming the reward $r_t$ has the following form:

$$r_t = \phi_t^\top \mathbf{o} + \phi_t^\top \mathbf{A} \phi_t \tag{12}$$

where $\{\phi_t, \mathbf{o}\} \in \mathbb{R}^{z \times 1}$, and $\mathbf{A} \in \mathbb{R}^{z \times z}$ and both $\mathbf{o}$ and $\mathbf{A}$ are learnable parameters. Following from the definition of the state-action value function $Q(s, a)$, the adjusted reward function can be substituted to yield:

$$Q^\pi(s, a) = \mathbb{E}^\pi[r_{t+1} + \gamma r_{t+2} + \ldots | S_t = s, A_t = a] \tag{13}$$

$$= \mathbb{E}^\pi[\phi_{t+1}^\top \mathbf{o} + \phi_{t+1}^\top \mathbf{A} \phi_{t+1} + \gamma \phi_{t+2}^\top \mathbf{o} + \gamma \phi_{t+2}^\top \mathbf{A} \phi_{t+2} + \ldots | S_t = s, A_t = a] \tag{14}$$

Dropping the conditional portion of the expectation for brevity, linearity of expectation can be used to split apart the terms containing $\mathbf{A}$ and $\mathbf{o}$. Then $\mathbf{o}$ is pulled out from the first term:

$$= \mathbb{E}^\pi[\phi_{t+1}^\top \mathbf{o} + \gamma \phi_{t+2}^\top \mathbf{o} + \ldots] + \mathbb{E}^\pi[\phi_{t+1}^\top \mathbf{A} \phi_{t+1} + \gamma \phi_{t+2}^T \mathbf{A} \phi_{t+2} + \ldots] \tag{15}$$

$$= \mathbb{E}^\pi[\phi_{t+1} + \gamma \phi_{t+2} + \ldots]^\top \mathbf{o} + \mathbb{E}^\pi[\phi_{t+1}^\top \mathbf{A} \phi_{t+1} + \gamma \phi_{t+2}^\top \mathbf{A} \phi_{t+2} + \ldots] \tag{16}$$

By recognizing the first expectation term as the successor features $\psi(s, a)$, Equation 16 can be rewritten as

$$= \psi^\pi(s, a)^\top \mathbf{o} + \mathbb{E}^\pi[\phi_{t+1}^\top \mathbf{A} \phi_{t+1} + \gamma \phi_{t+2}^\top \mathbf{A} \phi_{t+2} + \ldots] \tag{17}$$

Because $\phi^\top A \phi$ results in a scalar, the trace function $tr(\cdot)$ can be used inside the right-hand term:

$$= \psi^\pi(s, a)^\top \mathbf{o} + \mathbb{E}^\pi[\mathbf{tr}(\phi_{t+1}^\top \mathbf{A} \phi_{t+1}) + \mathbf{tr}(\gamma \phi_{t+2}^\top \mathbf{A} \phi_{t+2}) + \ldots] \tag{18}$$

By exploiting the fact that $\mathbf{tr}(\mathbf{AB}) = \mathbf{tr}(\mathbf{BA})$, the terms inside the trace function can be swapped to yield:

$$= \psi^\pi(s, a)^\top \mathbf{o} + \mathbb{E}^\pi[\mathbf{tr}(\mathbf{A}\phi_{t+1}\phi_{t+1}^\top) + \mathbf{tr}(\gamma \mathbf{A}\phi_{t+2}\phi_{t+2}^\top) + \ldots] \tag{19}$$

Because both $tr(\cdot)$ and $\mathbf{A}$ are linear, they can be pulled out of the expectation, giving:

$$= \psi^\pi(s, a)^\top \mathbf{o} + \mathbf{tr}(\mathbb{E}^\pi[\mathbf{A}\phi_{t+1}\phi_{t+1}^\top + \gamma \mathbf{A}\phi_{t+2}\phi_{t+2}^\top + \ldots]) \tag{20}$$

$$= \psi^\pi(s, a)^\top \mathbf{o} + \mathbf{tr}(\mathbf{A}\mathbb{E}^\pi[\phi_{t+1}\phi_{t+1}^\top + \gamma \phi_{t+2}\phi_{t+2}^\top + \ldots]) \tag{21}$$

Finally, the remaining expectation can be expressed as a function:

$$Q^\pi(s, a) = \psi^\pi(s, a)^\top \mathbf{o} + \beta \mathbf{tr}(\mathbf{A}\Lambda^\pi(s, a)) \tag{22}$$

$\beta \in \{0, 1\}$ is a hyperparameter that controls the inclusion of the non-linear component. We define $\psi^\pi$ and $\Lambda^\pi$ as:

$$\psi^\pi(s, a) = \mathbb{E}^\pi[\phi_{t+1} + \gamma \psi(s_{t+1}, \pi(s_{t+1})) | S_t = s, A_t = a] \tag{23}$$

$$\Lambda^\pi(s, a) = \mathbb{E}^\pi[\phi_{t+1}\phi_{t+1}^\top + \gamma \Lambda(s_{t+1}, \pi(s_{t+1})) | S_t = s, A_t = a] \tag{24}$$

## B ENVIRONMENTS

### B.1 AXES

In the Axes environment, each action moves the agent by 0.01 units in the desired direction with the traversable space defined by a $15 \times 15$ unit box centered at the origin $(0, 0)$. Within this environment eight separate goal locations exist split between train and test distributions. An episode ends when either the agent reaches the goal or more than 225 steps have elapsed. The agent's starting location is randomly sampled from a grid of $3 \times 3$ step units, centered at $(0, 0)$.

### B.2 REACHER

In the Reacher environment, an episode ends when 150 steps have elapsed or the agent is within 7cm of the goal.

Because the models can be used only with discrete actions, it was necessary to transform the environmental actions. Therefore, the four-dimensional continuous action space $\mathcal{A}$ was discretized using two values per dimension: the maximum positive and maximum negative torque for each actuator. An all-zero option was included that applies zero torque along all actuators, resulting in a total of nine discrete actions.

## C EXPERIMENTS

We use the factorization of matrix $\mathbf{A}$ with the two separate matrices, $\mathbf{L}$ and $\mathbf{R}$, with an inner dimension equal to $f = \frac{z}{2} - 1$. All models, unless specified otherwise, were trained using a synchronous version of A3C (Mnih et al., 2016). We trained all components on $n$-step trajectories, with $n = 5$, generated by 16 separate threads. The targets for $\psi$ and $\Lambda$ were estimated using an n-step return across latent states $\phi$. All models used an annealed $\epsilon$-greedy method for exploration. Within both the Axes and Reacher environment, $\epsilon$-greedy was annealed from 1.0 to a final over the first 250k steps.

In the Axes and Reacher environments the encoder and decoder each respectively contain one and two hidden layers with an embedding size equal to the double the raw state size. Both $\psi$ and $\Lambda$ increase the hidden dimension $z$ by a fixed factor before output. This factor $z_{\text{factor}}$ depends on the environment. All environments used a discount factor of $\gamma = 0.99$, $\lambda = 0.1$, and updated the parameters every $25k$ steps.

### C.1 AXES

We used an embedding size of $z = 8$ in the Axes environment. The final value used in $\epsilon$-greedy was 0.1 with a learning rate of $\alpha = 2.5e - 4$ was used.

### C.2 REACHER

We used an embedding size of $z = 12$ in the Reacher environment. The final value used in $\epsilon$-greedy was 0.05 with a learning rate of $\alpha = 5e - 4$ was used.

## D TRANSFER

During transfer we reinitialize the reward specific parameters $\mathbf{A}$ and $\mathbf{o}$ using an orthogonal initialization with a gain of 1. All other model parameters are held frozen and do not change. The learning rate is increased by a factor of $2\times$ and $10\times$ on the Axes and Reacher environments respectively. We anneal the exploration parameter from 1 to 0.1 in Axes and to 0.05 on the Reacher environments over the first 200k steps.

## E ADDITIONAL EXPERIMENTS

In this section we discuss additional experiments performed within the Axes environment. The Axes environment was used as it has simple dynamics and representation such that the resulting

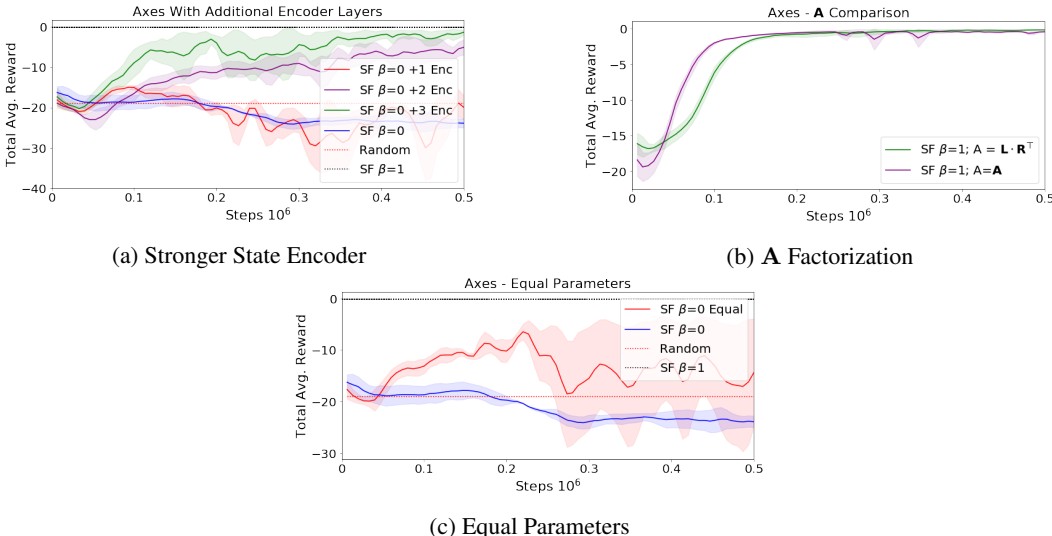

(a) Stronger State Encoder

(b) **A** Factorization

(c) Equal Parameters

Figure 7: Additional Experiments

performance is This primary motivation was to understand if there are any confounding factors in the performance of the linear and non-linear models.

### E.1 STRONGER STATE ENCODER

In this experiment we add additional fully connected layers to the encoder of the linear model. From Figure 7a we see that as the strength of the encoder increases so does the model's overall performance. This is unsurprising as the encoder is able to learn a better encoding of the state for reward prediction with additional parameters. To near the performance of the non-linear model requires three additional, four total, fully connected layers each with $z$ hidden units. This is a significant increase in parameters when compared to the non-linear variant which only has a single fully connected layer in its encoder. Even with the additional layers the linear version does not match the performance of the non-linear model.

### E.2 **A** FACTORIZATION

In this experiment we show the factorization of the parameter **A** has little effect on the model's performance. We examined using a full matrix of parameters, such that $\mathbf{A} \in \mathbb{R}^{z \times z}$ and a factorization where $\mathbf{A} = \mathbf{L} \cdot \mathbf{R}^\top$ where $\{\mathbf{L}, \mathbf{R}\} \in \mathbb{R}^{z \times f}$. From Figure 7b we see that the final performance of both methods are identical and they roughly converge at the same speed. The full matrix variant converges faster than the factorized version, but has nearly double the number of parameters. Learning the full matrix variant is a tractable learning problem and learning matrices of such scale has been done in other work, such those in state-transition models (Oh et al., 2015; Farquhar et al., 2017; Tasfi & Capretz, 2018). In the case of our model, we chose the factorized version to reduce the total number of parameters.

### E.3 EQUAL PARAMETERS

To ensure that the non-linear model's performance is not simply due to a greater number of parameters, we examine a linear variant with an approximately equal number of parameters. The parameters of the linear variant are increased by adjusting the number of hidden units, controlled by the hyperparameter $z$. From Figure 7c, we see that setting the number of parameters roughly equal to that of the non-linear model does help performance slightly but the linear model still cannot solve the task. We conclude that the new reward structure introduced by our paper, with its ability to model a non-linear reward, is the primary reason for the increase in performance.

