# OpenReview forum: "Non-Linear Rewards For Successor Features"
_ICLR.cc/2021/Conference — Reject_

### Official Review · AnonReviewer3 · 2020-10-27
**Proposes to train successor features to predict [phi, phi^2] rather than only [phi], but does not demonstrate clear benefit**

**Rating:** 4
**Confidence:** 4

**Review:**

This paper proposes to extend successful features by learning the second moments of cumulants in addition to the cumulants. They demonstrate that the resulting method performs better on 2D and 3D goal-reaching tasks (without obstacles) when the reward is the squared distance.

The paper is clearly written and extending successor features to non-linear rewards is an interesting problem. However, the framework does not so much provide a “novel formulation of successor features” but rather presents a specific instantiation. While this maybe be an interesting observation, the resulting experiments fail to compare to more appropriate baselines and are tested on rather simple domains that can be solved by a quadratic policy that is greedy with respect to the reward.

In detail:

The paper is effectively an instantiation of the successor feature that adds structure to the cumulants. In particular, it adds features that are products of other features, i.e. phi_new = [phi_1, phi_2, phi1 * phi_2, etc.]. Thus, the claim that this is a “novel formulation of successor features with a non-linear reward” seem inaccurate since the original successor feature formulation already handles the case where the reward is a non-linear function of the state.

Given the goal-directed nature of the tasks, the authors should compare to goal-conditioned methods [1,2,3] or at least demonstrate the method on tasks that cannot be solved by goal-conditioned methods. It is also an overstatement to say that the 3D reaching task is a particularly “difficult control task” given that much more challenging control tasks have been solved [4].


Minor comments:

Is it actually torque control? If so, how is it possible for the policy to learn given that it only has the XY location of the end effector? I’m under the impression that the meta-world tasks use end-effector velocity control.

“as the dimensionality of z grows, the number of parameters needed by Lambda grows exponentially” Do you mean that as the dimensionality of phi grows (or as z grows) the dimensionality of Lambda grows quadratically?

[1] Schaul, Tom, et al. "Universal value function approximators." International conference on machine learning. 2015.
[2] Andrychowicz, Marcin, et al. "Hindsight experience replay." Advances in neural information processing systems. 2017.
[3] Nair, Ashvin V., et al. "Visual reinforcement learning with imagined goals." Advances in Neural Information Processing Systems. 2018.
[4] Plappert, Matthias, et al. "Multi-goal reinforcement learning: Challenging robotics environments and request for research." arXiv preprint arXiv:1802.09464 (2018).

--- Post Rebuttal ---

I've read the author response. However, I do not plan on changing my score as my main concerns have not be addressed. In particular:

> the framework does not so much provide a “novel formulation of successor features” but rather presents a specific instantiation... Thus, the claim that this is a “novel formulation of successor features with a non-linear reward” seem inaccurate since the original successor feature formulation already handles the case where the reward is a non-linear function of the state.

The author response is

> There is no guarantee that the learned state features are able to find appropriate values. In theory yes, we agree but practically we found this not to be the case if we explicitly test for it. As our environments do, where the reward is a non-linear function of the state.

I understand that perhaps existing methods are not capable of learning good state features, and presenting a method for finding better state features (not the weights) would be interesting. However, the current paper simply hard-codes good features, which I do not find compelling.

---

> ### Author Response · Authors · 2020-11-23
> **.**
>
> Thank you for taking the time to review our paper and provide helpful comments.
>
> >original successor feature formulation already handles the case where the reward is a non-linear function of the state.
>
> There is no guarantee that the *learned* state features are able to find appropriate values. In theory yes, we agree but practically we found this not to be the case if we explicitly test for it. As our environments do, where the reward is a non-linear function of the state.
>
> > Given the goal-directed nature of the tasks, the authors should compare to goal-conditioned methods [1,2,3] or at least demonstrate the method on tasks that cannot be solved by goal-conditioned methods
>
> Our work used similar environments and baselines to that of related work in the SF subarea. We feel including baselines, such as Hindsight experience replay,  are outside of the scope of this work and all related SF papers.
>
> > It is also an overstatement to say that the 3D reaching task is a particularly “difficult control task”
>
> We agree, that if all methods under the umbrella of reinforcement learning are included, it can be taken as an overstatement. However, in comparison to related work in SF (eg. Barreto et al and Ma et al) it is an appropriately difficult task.
>
> > Is it actually torque control?
>
> Yes, it is torque control (See Section 3.1 where we describe the environment).
>
> > If so, how is it possible for the policy to learn given that it only has the XY location of the end effector?
>
> It has access to XYZ location of the end effector *and* the current goal (See Section 3.1 where we define the state).
>
> > I’m under the impression that the meta-world tasks use end-effector velocity control.
>
> The original version does, however we use a modified variant, which only uses the mujoco model definition.
>
> > “as the dimensionality of z grows, the number of parameters needed by Lambda grows exponentially” Do you mean that as the dimensionality of phi grows (or as z grows) the dimensionality of Lambda grows quadratically?
>
> Yes, thank you! We fixed this.

---

### Official Review · AnonReviewer4 · 2020-10-28
**An interesting generalization of successor features that requires a bit more development**

**Rating:** 4
**Confidence:** 4

**Review:**

This paper introduces a quadratic reformulation of successor features (SF), in which rewards are given by $r=\phi^\top \mathbf{o} + \phi^\top A \phi$ instead of the usual $r = \phi^\top w$. This generalization leads to learning a second-order term $\Lambda = \mathbb{E}[\sum_{t} \gamma^t \phi^\top \phi]$ to augment the expected featurization $\psi = \mathbb{E}[\sum_{t} \gamma^t \phi]$.

While this is a more expressive parametrization, learning $\Lambda$ is difficult due to its dimensionality. One of the strengths of SF is that it turns a high-dimensional prediction problem into a set of independent scalar Q-learning problems; incorporating $\Lambda$ arguably misses out on that important strength by no longer being able to treat the dimensions of the state featurization as independent. It seems that the learning problem can be made tractable by a low-rank factorization, but this does somewhat call into question whether a quadratic form is really the benefit or just expressivity from having more parameters. This could be answered by an experiment controlling for the number of parameters in $\phi$ (for the linear SF baseline) to match that in the quadratic variant.

The addition of $\Lambda$ is motivated by the limitation of a linear reward model:
> *The factorization follows from the assumption that reward can be predicted as the dot product between a state representation vector and a learned reward vector.*

This is not exactly an assumption as stated, since the featurization could be the reward itself ($\phi(s) = r(s), w = 1$), meaning that this decomposition is possible for any reward function. (See the discussion under Equation 2 in [Barreto, 2016].) That is not to say there is no limitation from linearity: it is true that for a *given* featurization, it may not be possible to linearly solve for any reward function, and that it may not be possible to find any suitable featurization for a set of many reward functions, but this is not discussed in enough depth to know exactly what issue is being referenced.

The experimental evaluation shows that the quadratic variant outperforms linear SF in two toy tasks. This is explained by the fact that these environments require a nonlinear model:
> *Finding a solution to the environmental reward structure is difficult as the reward is a
non-linear function of the features; in this case, the agent’s coordinates and the current goal location.*

However, even in the case of SF, the reward should not depend linearly on the observations themselves, but on a featurization of the observations. This is why a linear parametrization is suitable for any single reward function: the featurization can be arbitrarily complicated. Are you using observations (coordinates and goal location) instead of $\phi$ here?

Other experiments focus on exploration by adding noise to $\Lambda$ (instead of actions) and the structure of the learned $\Lambda$. The exploration idea is interesting, but somewhat underdeveloped, with there not being much justification and no clear empirical win. For what it is worth, the two environments studied might be simple enough that naive $\epsilon$-greedy exploration is good enough, and it is difficult to squeeze out much improvement without considering an environment which poses more of an exploration challenge. I did not understand the $\Lambda$ visualization in figure 5.

Though the major comparison of interest is between SF and the quadratic variant, it would be nice to explore a few more ablations of the method (like controlling for number of parameters, discussed above). It would also relieve a little bit of concern to have one or two more baselines on the standardized environments; the results in figure 3 look surprisingly sample-inefficient compared to modern Q-learning methods, so it could be worth figuring out why or showing that this is not the case by including another baseline.

**Questions:**
1. Where exactly does $\beta$ come from? It seems to appear between Equations 21 and 22 in Appendix A, but I cannot find the reason. Can the constant just be subsumed into $A$ in the reward regression in Equation 8? Is there ever a reason to set $\beta$ to anything besides 0 or 1?
2. What happens when you remove the reconstruction objective on $\phi$? Another advantage of the SF decomposition is that it can learn features informative for predicting values without having to rely on reconstruction, so can discard irrelevant parts of the state which would normally thwart a reconstruction objective. Incorporating a reconstruction term seems to miss out on that benefit. I realize prior work like [Kulkarni 2016] has used a similar auxiliary objective, so this isn't really a negative so much as a question as to why you think it is necessary.

**Copy-editing:**
1. Section 1.1 *The object* —> *the objective*

**Summary:**
Though this is an interesting generalization of SF, it does not seem quite ready for publication. I encourage the authors to more precisely motivate the quadratic form, since even after reading this paper its disadvantages (can no longer treat the dimensions of $\phi$ as independent, intractable due to dimensionality so requires approximation anyway) seem to outweigh potential advantages due to the increased expressivity (especially given the note above about a linear model being sufficient for any single reward).

---

> ### Author Response · Authors · 2020-11-23
> **.**
>
> Thank you for your review, we appreciate your comments, and feel out paper is better as a result. We have made several changes to our paper as a result of your comments (S2+S3+Appdx).
>
> > While this is a more expressive parametrization, learning Λ is difficult due to its dimensionality.
>
> Several works have learned much larger parametrized functions within RL, state-transition models being an example which often have several z x z matrices. As shown by our work we had little trouble learning this function.
>
> > by no longer being able to treat the dimensions of the state featurization as independent
>
> Could you explain this further? The original formulation does not explicitly guarantee this and if one were to use a learned encoder the state featurization would most likely end up being dependent on features from earlier layers.
>
> > It seems that the learning problem can be made tractable by a low-rank factorization
>
> The factorization was done to reduce parameters only. Our model still learns fine with the full rank matrix. We have added experimental support to show that both work fine in the Appendix. (see Appx. E.2).
>
> > This could be answered by an experiment controlling for the number of parameters in ϕ...
>
> We added a control in the Appendix (see Appx. E.3) where we set the parameters roughly equal: it does not cause equal performance.
>
> > This is not exactly an assumption as stated, since the featurization could be the reward itself...but this is not discussed in enough depth to know exactly what issue is being referenced.
>
> We have added further detail throughout the paper to improve clarity on our assumptions.
>
> > in two toy tasks
>
> We agree that the Axes environment is a toy task, but this was done by design so we can examine the learned functions easily (eg as done in S3.3+). The reacher task is a non-trivial task which requires control of 4 motors to move through 3d space.
>
> > this is why a linear parametrization is suitable for any single reward function: the featurization can be arbitrarily complicated. Are you using observations (coordinates and goal location) instead of ϕ here?
>
> We are using a learned featurization of the raw state. We had this mentioned in S3 but expanded further in our updated version.
>
> Further, we added another experiment in the appendix that shows the linear version does start to approach the non-linear versions performance but only if the encoder becomes significantly stronger (eg. +1->+3 layers). See Appx E.1.
>
> The non-linear and linear variants differ only by the inclusion of the lambda function (setting beta=1 or 0)
>
> >Other experiments focus on exploration by adding noise to Λ (instead of actions) and the structure of the learned Λ. The exploration idea is interesting, but somewhat underdeveloped
>
> We adjusted to text to make it clearer that these are preliminary results and used it to develop some intuition to what the new quantities have learned.
>
> > with there not being much justification
>
> Added further justification in the text.
>
> > no clear empirical win.
>
> We see it as an interesting alternative. In future work we hope to develop the idea further, which will hopefully result in a strong empirical win.
>
> > I did not understand the Λ visualization in figure 5.
>
> This is similar to the visualizations done in Barreto et al. and Hansen et al. (from our refs). It shows a variance like quantity over the state of the environment.
>
> > it would be nice to explore a few more ablations of the method
>
> We have added a new Appendix section as a result of your comments. See Appendix E.
>
> > the results in fig 3 look surprisingly sample-inefficient compared to modern Q-learning methods
>
> When compared to the meta-world paper, which includes a similar reacher environment, they require ~5x more samples for policy gradient type methods. We believe the 0.5e6 samples are not quite out of line. Also, we include many more samples after convergence to show the methods stability.
>
> > Where exactly does β come from? It seems to appear between Equations 21 and 22 in Appendix A, but I cannot find the reason. Can the constant just be subsumed into A in the reward regression in Equation 8? Is there ever a reason to set β to anything besides 0 or 1?
>
> We introduced it as an easy way to refer to the linear and non-linear model.
> Yes, it can be added into Equation 8.
> No, there is no reason to set it to any other value besides 0 or 1.
>
>
> > What happens when you remove the reconstruction objective on ϕ?  Another advantage of the SF decomposition...state which would normally thwart a reconstruction objective.
>
> We found that performance went down as we needed something "force" it to learn useful features (zero is a fixed point).
> We believe that this is the main motivation behind using a reconstruction objective, though others could probably be valid (predict next frame, contrastive loss, etc.)

---

### Official Review · AnonReviewer2 · 2020-10-29
**The natural extension of successor features to 2nd order, but let down by experimental section**

**Rating:** 4
**Confidence:** 4

**Review:**

Successor representations are an old idea that has seem recent interest in the ML community. The idea is conceptually straightforward, by assuming the rewards are linear in some space $r = \vect{\phi}(s, a) \cdot \vect{w}$ then we learn something analogous to an action-value function for the discounted expected features under a policy so that the action-value on task $\vect{w}$ is $Q^\pi_{\vect{w}}(s, a) = \psi^\pi(s, a) \cdot \vect{w}$. This allows computing the action value for the policy under a new task $\vect{w}'$ straight.

One limitation is the assumption that there is some feature space where the rewards of all tasks can be linear in this space.

The key idea is to extend the idea of successor features to the 2nd order relation between features and reward so now the assumption is the reward for all tasks is defined by eq 1 in the paper
so in addition to computing the expected features in the future $\psi^\pi(s, a)$ (a vector value) the autocorrelation of the features $\Lambda^\pi(s, a)$ must also be track (a matrix).

They construct a method for learning features as an autoencoder of the state and experiment on some simple tasks such as reaching to different locations comparing this ``second order'' method against successor features.

Strengths:
- This does seem like the natural ``second order'' version of successor features. I'm not aware of any closely related prior work to extend successor features in this way and it clearly allows for a more expressive description of rewards.

- Paper is mostly well-written and communicated, including try to help develop an intuition for the new quantities introduced.

- The interpretations of $\Lambda$ are interesting and ideas around how to use it for exploration are intriguing.

Weakness:

The primary weakness of this work is the experimental section. There seems to a number of different issues all conflated into one set of experiments.

Firstly, whether using successor features (SF) or 2nd order successor features (SF^2), $\psi^\pi$ and $\Lambda^\pi$ are a function of a particular policy. It is unclear if, for each training task a different $\mathbf{w} = \mathbf{o} + \mathbf{A}$ was learned (as one might expect in a SF setup), there is never any indices indicating these are task specific so perhaps there is only one version of these learned for all training tasks? It seems that only a single policy is learned for all training tasks. The setup should be clarified and the reasoning for these choices explained.

This means that over both training and test tasks only a single policy is available. Therefore, any hope of solving an individual task must be due to computing the advantage under this policy which is quite limiting. Both Barreto et al., and Ma et al. estimate the successor features for a set of policy (indeed the main contribution of Barreto is to show how to use Generalized Policy Improvement (GPI) to construct a policy for a new task defined by $w'$ from a set of existing policies, typically one per training task, there seems no reason GPI cannot be used here).

Finally, it would be helpful when introducing a conceptual idea as here to have a simple version where only that idea is needed. In this case, by (as in Barreto) experiments with fixed, pre-defined features. Here, in both experiments the features are learned using an autoencoder of the state space (which is not guaranteed to learn a representation which is ideal for use in successor features).

These limitations make it hard to interpret the experiments or compare with other work. For example, the two tasks appear quite similar to Barreto et al., yet here the SF baseline ($\beta=0$ in the paper) performs worse than random, even on the baselines. It is hard to interpret why this: is it due to the learned features not being good for this task, using only one policy for all tasks or the non-linearity of reward? It seems clear that e.g. for the Axes task if there was a feature for being located on each potential goal and a policy per task then, at least on the training tasks, SF should perform much better than random (by learning to reach for the targets).

Ideally, SF would also be compared against other methods for transfer such as meta-learning (e.g MAML). The existence of such methods, along with model-based approaches such as used in Go mentioned in the introduction probably means the claim in the introduction that "current algorithms cannot transfer a learned policy between related tasks" is too strong.

This issues are fixable by more careful experimentation and clarity on the exact setup of the problem. As part of that the captions for figures 3 and 4 could be extended.

A more general weakness is that, while this paper improves the expressiveness of SF by allowing a 2nd order relationship between features and reward, it is not clear if this is the key limitation of successor features. Namely, it still does not allow transfer between tasks where the transition function has changed ([1] should probably be cited as an attempt to extend SF in this direction) and only supports estimating the feature occupancy under existing policies. I personally would regard these are the more limiting factors in SF rather than the linearity of reward. These issues should be discussed and the limitations and weakness compared to e.g. meta-learning approaches should be discussed (and as mentioned above, ideally compared).

Just to be explicitly clear the rating given to the paper is the paper in it's current state. I think if the issues above are addressed this is an interesting paper and I would rate higher.

[1] Zhang, Jingwei, et al. "Deep reinforcement learning with successor features for navigation across similar environments." 2017 IEEE/RSJ International Conference on Intelligent Robots and Systems (IROS). IEEE, 2017.

---

> ### Author Response · Authors · 2020-11-23
> **.**
>
> Thank you for your review, we appreciate your comments, and feel out paper is better as a result. We have made several changes to our paper as a result of your comments (S2 and S3).
>
>
> > there is never any indices...the setup should be clarified and the reasoning for these choices explained.
>
> There is one policy/SF learned over all tasks in the training distribution. During transfer we hold everything frozen and retrain just the reward components.
>
> We added additional detail to section 3 to clarify this further.
>
>
> >Therefore, any hope of solving an individual task must be due to computing the advantage under this policy which is quite limiting
>
> Could you clarify this further? The reward components can “induce” different policies and as long as the environmental structure remains the same, as assumed by SFs, then there should be no loss of generality. The model as is has no issue transferring to a group of related tasks or singular ones.
>
> > Finally, it would be helpful when introducing a conceptual idea as here to have a simple version where only that idea is needed.
>
> We feel that the Axes environment does this, there is very little overhead/confounding factors. Additional detail was added to the environment description to help motivate its use.
>
> > Here, in both experiments the features are learned using an autoencoder of the state space
>
> Initially in our development of the paper we used the raw features on the Axes environment. The linear variant did much worse. With the learned state features it at least has *some* chance of learning a possible state representation.
>
> > using an autoencoder of the state space … which is not guaranteed to learn a representation which is ideal for use in successor features
>
> This is exactly the central assumption our work builds upon, which is why we believe the Axes is a perfect base environment.
>
> > For example, the two tasks appear quite similar to Barreto et al., yet here the SF baseline (β=0 in the paper) performs worse than random, even on the baselines.
>
> They are similar but not exact, in particular how the state is represented. In other SF papers each state feature would be the computed distance to one of N goal locations (eg. feature 0 of \phi^(0)_t would be the distance from the agent to the 0th goal) so the reward component must simple be a 1-hot encoding of the current active goal.
>
> Our state space instead provides the raw coordinates of the agent and the current goal. The reward component must model the negative euclidean distance between the goal and agent, which is a non-linear function of the raw state. The baseline, with linear reward model, cannot do this.
>
> As for why it does worse than random: we believe that the agent eventually learns a poor state/reward representation that creates a degenerate policy (eg. always go up and left or move in a circle).
>
> (See S3 and the appendix for further details)
>
> > Ideally, SF would also be compared against other methods for transfer such as meta-learning (e.g MAML).
>
> We feel that this is out of the scope of what we wished to focus on within this paper. Especially, given the related literature in the sub-area of SFs (environments used, baselines, etc.).
>
> >the claim in the introduction that "current algorithms cannot transfer a learned policy between related tasks" is too strong.
>
> Agreed, the claim was much too strong. We have adjusted the paper and reduced the strength of our claim to be less absolute.
>
> > This issues are fixable by more careful experimentation and clarity on the exact setup of the problem.
>
> We have added additional details to the experimental section that should further improve clarity.
>
> > Captions for figures 3 and 4 could be extended.
>
> Done.
>
> > …it is not clear if this is the key limitation of successor features.
>
> We feel that this assumption is a weakness in the framework itself, as it flows from the assumption that the reward can be predict from a linear combination of the features.
>
> > Namely, it still does not allow transfer between tasks where the transition function has changed ... and only supports estimating the feature occupancy under existing policies.
>
> Yes, we agree that this is indeed a weakness of the SF in general as it can only transfer between related tasks iff the environment structure is the same.
>
> We added a paragraph discussion the weaknesses, including the relevant paper you provided, at the start of section 2.
>
> > it is not clear if this is the key limitation of successor features.
>
>
> We feel that this is indeed a key limitation of SF, it directly stems from the assumption that the reward can be predicted linearly from the state features. As you pointed out, “…which is not guaranteed to learn a representation which is ideal for use in successor features….”, there is no guarantee in practice that the ideal features can be learned from just the encoder.
>
> This is directly tested for in all our experiments: the reward is a non-linear function of the state features.

---

### Official Review · AnonReviewer1 · 2020-10-30
**Sound extension to Successor Features, but empirical implications unclear**

**Rating:** 4
**Confidence:** 5

**Review:**

The authors extend the Successor Features (SF) framework to allow for a quadratic  relationship between the features and rewards, rather than the strictly linear one given in the initial formulation. The derivation is relatively clear and appears to be correct, with the nice result being that the additional term needed to account for the non-linearity can still be estimated via a Bellman equation.

One fact that the authors fail to address is that the linearity between the features and rewards doesn't limit the expressiveness of value functions computed by SF, since the features can be an arbitrarily non-linear function of the observations. Now, it might be the case that the ability to generalize to novel tasks is greater when making the rewards be non-linear in the features, but the author should make this distinction.

Indeed, this ties into my larger issue when the paper: the Axes task (and maybe the Reacher task depending on the dynamics) is very simple, to the point that it is very unclear why the linear SF formulation does worse than not only the quadratic case, but than a random agent. As per my argument about expressivity, both SF agents should be equally able to solve the training tasks, with the only space for an advantage for the non-linear version coming from generalization.

But the Axes task appears too simple to even be used for comparing generalization performance (unless I'm mistaken about the dynamics). If the tasks terminate upon reaching the goal, and the start states are all in the middle of the goals, then couldn't an agent linear in the *observations* solve the tasks optimally? e.g. map the difference between the current and goal observations to the action most closely corresponding to the resulting direction.

The Reacher task might also suffer from these problems, but I'm unsure without further details. Do the actions act on the torques of the robot motors, or do they directly impact end-effector space? If it is the latter, then Reacher appears to just be Axes with an addition spatial dimension, which would still be readily solved without any non-linear function approximation.

I apologize if I sound overly harsh. The extension is interesting, but currently its unclear if these experiments provide any support to your claims about its advantages. Unless I'm mistaken about the dynamics of these environments, I'd suggest showing results on tasks of known complexity, like the Doom game Deep SR uses or the Scavenger environment from the SF paper.

The exploration angle is quite interesting, but currently its hard to understand the motivation. Why is perturbing this expected variance-like object a good idea? Expanding upon that would be greatly appreciated, though for it to be the main claim of the paper there would also have to be some comparison to alternative exploration methods (e.g. Noisy Nets or Bootstrap DQN).

---

> ### Author Response · Authors · 2020-11-23
> **.**
>
> We would like to thank you for suggestions and feedback.  We have made a few changes to the paper as per your suggestions (S1-S3).
>
>
> > One fact that the authors fail to address is that the linearity between the features and rewards doesn't limit the expressiveness of value functions computed by SF, since the features can be an arbitrarily non-linear function of the observations.
>
> We added further clarification and a strong statement of our assumption throughout the paper (introduction and model sections in particular). There is no guarantee that the features learned by the encoder will be able to accurately predict the reward - this is what we have focused on. This assumption that we make is not in itself a strong one.
>
>
> > the Axes task … is very simple, to the point that it is very unclear why the linear SF formulation does worse than […] the quadratic case
>
> The linear variant *cannot* model the reward correctly, the negative euclidean distance, as it is a non-linear function of the current state. The linear model has no hope of modelling this function: how can it express a non-linear function such as a square power or a square root?
>
>
> We added more detail to the paper explaining this.
>
>
> > worse than the random agent.
>
> We believe that the linear agent learned a sub-optimal encoding of the state space which then causes a worse than random policy.
>
>
> We added more detail to the paper explaining why we believe its performance was worse.
>
>
> > then couldn't an agent linear in the observations solve the tasks optimally? e.g. map the difference between the current and goal observations to the action most closely corresponding to the resulting direction.
>
> No, it cannot. See our previous answer on this topic. In the case of the Axes environment even if it learned to take the difference between the goal (g) and agent (a) x and y positions, without a non-linearity (eg. absolute or square power), it has no way of knowing *which* way to go. Eg. the best it could do might be something like: g_x - a_x + g_y - a_y.
>
>
> > The Reacher task might also suffer from these problems, but I'm unsure without further details.
>
> Additional details are in the appendix. We did however add a strong explicit mention of the fact the agent controls the torque applied to 4 motors. We believe this is a non-trivial problem.
>
>
>
> > Do the actions act on the torques of the robot motors, or do they directly impact end-effector space?
>
> The agent controls the torque applied to 4 motors.
>
> > I apologize if I sound overly harsh.
>
> No offence taken. We really do appreciate the feedback and feel the paper has become much clearer from your suggestions+questions.
>
> >The exploration angle is quite interesting, but currently its hard to understand the motivation. Why is perturbing this expected variance-like object a good idea? Expanding upon that would be greatly
> appreciated…
>
> We added further detail and our reasoning behind this in the text.
>
> > For it to be the main claim of the paper there would also have to be some comparison to alternative exploration methods (e.g. Noisy Nets or Bootstrap DQN).
>
> We dialled back the claim as this was a preliminary examination of the function and hope to develop the intuition+applications further in future work. The inclusion was primarily because we found it interesting and wanted to develop intuition for the newly introduced quantity.

---

> > ### Comment · AnonReviewer1 · 2020-11-24
> > **Re: limitations of rewards being linear in features**
> >
> > "The linear variant cannot model the reward correctly, the negative euclidean distance, as it is a non-linear function of the current state. The linear model has no hope of modelling this function: how can it express a non-linear function such as a square power or a square root?"
> >
> > Again, the mapping between observations and features can be arbitrarily non-linear, so a model linear in such features can model *arbitrarily non-linear* reward functions. And this easy to do in practice; just train a neural net to predict the reward and then treat the last layer as the features.
> >
> > "We believe that the linear agent learned a sub-optimal encoding of the state space which then causes a worse than random policy."
> >
> > This just shows that you have a very poor baseline. My hypothesis is that it's your architecture; why not just make phi be learned on the basis of reward prediction (Figure 1B)? Forcing Psi and reconstruction (Figure 1A) to be functions of phi is unnecessary and appears to be greatly hurting performance. Even if the architecture is inherited from prior work, these assumptions should be rethought if you're doing worse than random on a trivial task.
> >
> > "No, it cannot. See our previous answer on this topic. In the case of the Axes environment even if it learned to take the difference between the goal (g) and agent (a) x and y positions, without a non-linearity (eg. absolute or square power), it has no way of knowing which way to go. Eg. the best it could do might be something like: g_x - a_x + g_y - a_y."
> >
> > This was about the optimal *policy* being linear in the observations, and I still believe this is the case. e.g. logits for the up action: g_y - a_y, right action g_x - a_y. This is largely an aside about the simplicity of the task rather than being directly relevant to the linear/non-linear successor features distinction.
> >
> > -----
> >
> > Thanks for clarifying the reacher task, actions being in torque-space does add some complexity. And I agree with the revised motivation that non-linear successor features could take some burden off of the feature encoder. But you have not demonstrated this to be the case in practice. The idea that a basic non-linear feature encoder couldn't render euclidean distance linear is dubious, and makes the possible effect on the harder task unclear.

---

> > > ### Author Response · Authors · 2020-11-24
> > > **.**
> > >
> > > > Again, the mapping between observations and features can be arbitrarily non-linear, so a model linear in such features can model arbitrarily non-linear reward functions. And this easy to do in practice; just train a neural net to predict the reward and then treat the last layer as the features.
> > >
> > > We are not debating this. Instead, we are focusing on and testing the case if the state features *are* suboptimal and cannot fully represent the reward as a linear function of state features.
> > >
> > > > This just shows that you have a very poor baseline.
> > >
> > > The architecture works fine if the encoder is much stronger or the state features are of similar design to other works (eg. the feature j is the distance to goal j so for that task w^(j) must be simply equal 1).
> > >
> > > Also note that the linear and non-linear variants are identical in every way except we simply set beta=1 for the non-linear variant.
> > >
> > > >why not just make phi be learned on the basis of reward prediction (Figure 1B)? Forcing Psi and reconstruction (Figure 1A) to be functions of phi is unnecessary and appears to be greatly hurting performance.
> > >
> > > You are correct, which is why we do not train through Psi. Phi is only learned on the basis of the reward prediction and reconstruction. As mentioned in the text we do not backprop psi, or Lambda in the case of beta=1, through the encoder. So both variants are optimizing the reward and state features in the exact same way.
> > >
> > >
> > > > This was about the optimal policy being linear in the observations, and I still believe this is the case. e.g. logits for the up action: g_y - a_y, right action g_x - a_y. This is largely an aside about the simplicity of the task rather than being directly relevant to the linear/non-linear successor features distinction.
> > >
> > > This is not possible for a single reward vector w. In the SF framework there is only one reward component, w, available over *all* actions (this is a core structural feature of the framework). It is not possible to have a w vector that can model what you've written above where you ignore part of the feature space in one action and not in the other. You would need one w per action for what you've written. Roughly speaking,
> > >
> > > psi_t = phi_t = [g_y, a_y, g_x, a_x ]  (assume gamma=0 for this example but still holds otherwise)
> > > eg. up w: [1, -1, 0, 0] and right w: [0, -1, 1, 0]. (or right w: [0, 0, 1, -1] assuming a_y was a typo)
> > >
> > > Further, while this is the optimal policy, w is learned only from predicting the reward r_t at timestep t. How could the linear variant find an optimal setting for a policy a suggested above, with one w, while predicting only the reward? (assuming either the state features are sub optimal or raw as shown above.)
> > >
> > > Also, the learned policy needs to perform well over a *distribution* of training tasks, placed in compass rose coords, with this singular w. The distribution of tasks most likely adds to its poor performance as it cannot settle on an appropriate policy/reward function that fits all (even less surprising it does worse than random).

---

### Decision · Program_Chairs · 2021-01-07
**Final Decision**

**Decision:**

Reject

**Comment:**

This paper extends the idea of successor representations. Typically the reward is compute linearly on top of states in this setting but the authors relax it to have a quadratic form.

${\bf Pros}$:
1. A novel formulation of the successor representation where the reward does not follow the linearity assumption
2. The idea of using a second order term for the reward branch is interesting and could have meaningful implications for learning and exploration.

${\bf Cons}$:
1. All authors agree that the experimental results do not clearly validate the advantage of this method. More work is needed to establish the effects of using this particular reward structure on a wide variety of tasks

2. Both R2 and R4 were unconvinced of the limitations of the linearity assumptions in the original successor representation formulation -- especially in the case when the state is represented by a non-linear function approximator.

The ideas presented in this paper are quite interesting and promising. But more experimental work is needed to show the benefits of this approach.